# Perception and willingness to accept COVID-19 Vaccines: A cross-sectional survey of the general population of Sokoto State, Nigeria

**Oche Mansur Oche[1,2]\*, Habibullah Adamu[1,2], Musa Yahaya[1], Hudu Garba Illo[2], Abdulaziz Mohammad Danmadami[2], Adamu Ijapa[2], Asmau Mohammad Wali[2], Hamza Yusuf[2], Hafsat Muhammad[2], Abba Aji[2]**

1 Department of Community Health, Usmanu Danfodiyo University, Sokoto, Nigeria, 2 Department of Community Medicine, Usmanu Danfodiyo University Teaching Hospital, Sokoto, Nigeria

\* ochedr@hotmail.com, oche.mansur@udusok.edu.ng

**Data Availability Statement:** All relevant data are within the paper and its Supporting Information files.

## Abstract

The number of confirmed cases of COVID-19 globally is well over 400 million, however, the number of cases is showing a downward trend especially in developed countries largely as a result of effective vaccination against COVID-19. In developing countries, vaccination coverage is still very low as a result of vaccine hesitancy, which could be attributed to misconceptions about COVID-19 itself and its newly developed vaccines. This study assessed COVID-19 vaccine acceptance and perception amongst the adult population in Sokoto state, Nigeria. A cross-sectional study was conducted in Sokoto state among 854 respondents selected via a multi-stage sampling technique. Data was collected electronically using a set of structured questionnaire and analysis was done using IBM SPSS version 25. Respondents' perception was assessed using a 5-point Likert scale ranging from strongly disagree (1) to strongly agree (5). Respondents having a score of 3 and below were graded as having poor perception and those having scores above 3 were graded as having good perception. Respondents' ages ranged from 17 to 76 years, with a mean of 34.8±12.07; more than half [474(53.7%)] of the respondents were males, 667(75.5%) were married and 539(61.0%) had formal education. The majority [839(95.0%)] of the respondents had a good perception of COVID-19 vaccine; 49.9% agreed enough research would be required on the safety of the vaccine. The majority, (72.4%) expressed their willingness to accept the COVID-19 vaccine (male 38.4% vs. female 34.0%); 410(47.4%) said they can spend more than one hour to get the vaccine. Significant predictors of willingness to accept COVID 19 vaccine include age (p = 0.006; aOR = 0.223; 95% CI = 0.077–0.645), education (p<0.001; aOR = 1.720; 95% CI = 1.274–2.321) and perception of COVID 19 vaccine (p<0.001; aOR = 0.020; 95% CI = 0.009–0.044). The majority of the respondents had a good perception of COVID-19 vaccine and more than two-thirds were willing to be vaccinated with the vaccine. Government should make the vaccine available for vaccination since a significant proportion of the respondents expressed their willingness to accept the vaccine

**Funding:** The authors of this paper received no specific funding for this work.

**Competing interests:** The authors have declared that no competing interests exist.

## Introduction

Corona Virus disease (COVID-19) is a viral pandemic that was discovered in China, in the year 2019. The origin of the viral disease was traced to a wet market in Wuhan, a Chinese City in Hubei province. The disease is caused by a novel virus called Severe acute respiratory syndrome coronavirus 2 (SARS-CoV-2) [1].

As of 10th February 2022, the global confirmed cases of COVID-19 stand at 402,044,502, with 5,770,023 deaths and the number keeps growing on a daily basis [2]. In Nigeria, 253,838 confirmed cases of COVID-19 were reported as of 8th February, 2022 with 3,139 deaths [3]. The COVID-19 pandemic has inflicted almost unimaginable harm on the life, health, and economy of many nations. Along with hygienic and behavioral control measures, vaccination is the most successful way of limiting or eliminating viral infection and spread. Even the best vaccine cannot be effective if it is not used. Recent surveys found that 50% of Americans said they are willing to take the vaccine, 30% are unsure, and 20% are refusing the vaccine [4]. In another survey of adult Americans, 58% intended to be vaccinated, 32% were not sure, and 11% did not intend to be vaccinated [5]. This number of participants is likely below the threshold needed for homogeneous herd immunity [6] and will leave many residents vulnerable to the disease, even with a vaccine available. Promoting the uptake of vaccines (particularly those against COVID-19) will require understanding whether people are willing to be vaccinated, the reasons why they are willing or unwilling to do so, and the most trusted sources of information in their decision-making [7,8]. Survey among the general public in African countries reported an acceptance rate of 81.6% in South Africa and 65.2% in Nigeria [9]. Early knowledge, attitudes and practices studies regarding COVID-19 from North-Central Nigeria reported an acceptance rate of 29.0%, which highlights the need for more studies for an accurate depiction of COVID-19 vaccine hesitancy in Africa due to possible large regional and sub-regional variations [10]. The World Health Organization (WHO) recommends a preemptive strategy to overcome vaccine hesitancy and build trust in a vaccine to prepare for maximum efficacy when a vaccine is available [11,12].

There are several COVID-19 candidate vaccines at various stages of development with few approved for use by regulatory authorities and WHO. The Pfizer/BoiNTech vaccine, Astrazeneca/University of Oxford vaccine, Sinovac vaccine, and Moderna COVID-19 vaccines are already approved for use by regulatory authorities in different countries across the globe [13]. Some of these vaccines are in activated viral particles while others are live attenuated viruses. There are some candidate vaccines that are viral subunits while few are mRNA vaccines. Some of these candidate vaccines are scheduled to be administered only once but the majority are to be administered twice (2doses) in other to achieve vaccine efficacy above 90%. About 84% of these vaccines are to be administered by injection with 76%, 5% and 3% through the intramuscular, intradermal and subcutaneous routes respectively. A few of the vaccines are to be administered by oral route [13].

Several groups of people and individuals across the globe have developed negative perceptions and misconceptions about the developed COVID-19 vaccines [14]. This has led to vaccine hesitancy among various populations with a reasonable proportion of the population not willing to be vaccinated with the newly developed vaccines due to perceived safety issues. In a broader context, vaccine hesitancy as defined by the WHO is the delay in acceptance or refusal of vaccination despite the availability of vaccination services [14]. The complex nature of motives behind vaccine hesitancy can be analyzed using the epidemiologic triad; environment, agent and host factors [15,16]. Environmental factors include public health policies, social factors and the messages spread by the media [17–19]. The agent (vaccine and disease) factors involve the perception of vaccine safety and effectiveness, besides the perceived susceptibility

to the disease [19–21]. Host factors are dependent on knowledge, previous experience, educational and income levels [16,22]. Previous studies have shown that vaccine hesitancy is a common phenomenon globally, with variability in the cited reasons behind the refusal of vaccine acceptance [23–25]. The most common reasons include perceived risks vs. benefits, certain religious beliefs and lack of knowledge and awareness [26,27]. The aforementioned reasons can be applied to COVID-19 vaccine hesitancy, as shown by recent publications that showed a strong correlation between intent to get coronavirus vaccines and its perceived safety [28]. Association of the negative attitude towards COVID-19 vaccines and unwillingness to get the vaccines, and the association of religiosity with lower intention to get COVID-19 vaccines [29]. To date, there has been no prior study among the general population of Sokoto State investigating their perceptions, acceptance and hesitancy to receive COVID-19 vaccine. This study, was therefore aimed at determining the perception and hesitancy of the adult population of Sokoto state towards COVID- 19 vaccines. It is hoped that findings will help further interventions aimed at increasing the acceptability of the vaccines.

## Materials and method

The study was conducted in Sokoto state in Northwestern part of Nigeria; it has 23 Local Government Areas (LGAs) comprising three senatorial zones namely Sokoto East, Sokoto North and Sokoto South senatorial zones respectively. Sokoto State has a population (projected) of 6,391,000 as of 2022 based on the 2006 general census [30].

The study utilized a cross- sectional design involving all adults residing or plying their trades in the state. Only those 18 years and above (male and female) resident within the study area, who have spent not less than six months in the study area were recruited into the study.

Using a prevalence of 66.1% for the proportion of parents who refused COVID- 19 vaccine in a previous study, a sample size of 344 was calculated using the Cochrane formula for estimating sample size in descriptive studies $n = z^2pq/d^2$ [31].

With an anticipated response rate of 95%, the sample size was adjusted to 362. However, since cluster sampling was used to select respondents, the sample size was further adjusted to 905 using a 2.5 design effect.

The sampling technique for Lots Quality Assurance Sampling (LQAS) was modified and adopted for the selection of study participants [32], who were selected via a four- stage sampling process. One LGA was selected from each of the 3 senatorial zones in the state by simple random sampling through balloting to yield a total of 3 LGAs. Three catchment areas (wards) were selected from each of the selected LGAs by simple random sampling through balloting to yield a total of 9 catchment areas. Five supervision areas (settlements) were selected from each of the selected catchment areas by simple random sampling through balloting to yield a total of 45 supervision areas (settlements). Nineteen (19) participants (males and females) were selected from each of the selected supervision areas (settlements) to obtain a total of 855 study participants.

Data was collected using a set of interviewer administered questionnaire which was prepared in English language and translated to the local language–Hausa through a two-way process to verify the accuracy of the translation by two Hausa scholars. The questionnaire comprised four sections; socio-demographic characteristics, perception, acceptance/hesitancy and willingness to be vaccinated. It was pretested in two selected communities in a different LGA outside the selected LGAs after which necessary amendments were made. The questionnaire was uploaded on Open Data Kit (ODK) software using android hand–held smart devices which were scripted to prevent or minimize data entry errors, ease timely data collection, ensured completeness of the information and subsequent processing and analysis.

Seven Resident Doctors of the Department of Community Medicine of Usmanu Danfodiyo University Teaching Hospital, (UDUTH) Sokoto were used as research assistants for the data collection. They were trained by the Researchers for two days; each training session lasted for 2 hours. The training covered an overview of COVID-19 and covid-19 vaccine, general principles of research, objectives of the study, conduct of research, interpersonal communication skills and administration of research instruments.

Data collected was transmitted daily and temporally stored in the kobo collect server, which also helped in the monitoring of data collection activities. Data was subsequently downloaded in excel format for analysis and later exported to IBM SPSS version 25 for analysis; data editing and cleaning were done using constraints, restrictions and required functions on ODK.

Respondents' perception was assessed using a 5-point Likert scale ranging from strongly disagree (1) to strongly agree (5). For each of the variables, the median score was calculated and graded as 1 = Strongly disagree, 2 = disagree, 3 = don't know, 4 = agree, and 5 = strongly agree. All the Likert type items for perception that were framed negatively were reversed so that each of the items follow the same direction (positive direction). Each positive response to a perception question was awarded a score of one mark while zero mark was awarded to each negative response; the scores were summarized as mean score ranging from 1–5. A score of 1–3 was graded as poor perception whereas scores of 4 and 5 were graded as good perception. Continuous variables were summarized as mean and standard deviation, whereas categorical variables were summarized as frequencies and percentages. The Chi-square test was performed to assess the existence of an association between categorical variables; the level of statistical significance was set at $p < 0.05$.

Ethical approval was sought and obtained from the Health Research Ethics Committee of Usmanu Danfodiyo University Teaching Hospital Sokoto and individual verbal informed consent was sought from the participants before administering the questionnaire to them.

## Results

The mean age of the respondents was 34.82 ± 12.06 years, the age group 20–29 years had the highest proportion [293 (33.2%)] of respondents. More than half of the respondents were males [474 (53.7%)], three-quarters of the respondents were married [667 (75.5%)] and the majority were Muslims [824 (93.3%)] and belonged to Hausa ethnic group 718 (81.3%). More than three- fifth of the respondents had formal education [539 (61.0%)] however, only a third of the respondents were gainfully employed [308 (34.9%)] (Table 1).

Regarding respondents' perception of the Covid-19 vaccine, about half [430(49.9%)] of them agreed enough research has been conducted on the safety of the vaccine, 302(35.1%) disagreed with the statement that Covid- 19 vaccine can lead to infertility. Regarding its effectiveness, 285(33.7%) agreed the vaccine is effective in stopping Covid-19; 139(16.2%) agreed that the side effects of the vaccine are worse than the Covid -19 disease itself (Table 2).

Fig 1 shows the perception of respondents regarding the Covid-19 vaccines where the majority of the respondents [785(89.4%)] had a poor perception regarding covid-19 vaccine.

Close to three-quarters [636(72.4%)] of the respondents would want to be vaccinated for COVID- 19, majority of the respondents [731(83.2%)] have never rejected vaccines for their children. Close to half [410(47.4%)] of the respondents were willing to travel more than one hour to get a vaccine, and more than half 487(55.4%) reported that they will accept the vaccine for their families if a new coronavirus (COVID-19) vaccine became available (Table 3).

Fig 2 depicts the willingness of respondents to accept the Covi-19 vaccines. Up to 636 (72.4%) of the respondents expressed their willingness to accept Covid -19 vaccine once it is made available [males 337(38.4%) vs. females 299(34.0%)] (Fig 2).

**Table 1. Sociodemographic characteristics of respondents.**

| Variable | Frequency (%) |
|---|---|
| **Age group (years)** | |
| <20 years | 41 (4.6) |
| 20–29 years | 293 (33.2) |
| 30–39 years | 274 (31.0) |
| 40–49 years | 157 (17.8) |
| 50–59 years | 63 (7.1) |
| ≥60 years | 55 (6.2) |
| **Mean ±SD** | **34.82 ± 12.068** |
| Sex | |
| Male | 474 (53.7) |
| Female | 409 (46.3) |
| **Marital status** | |
| Married | 667 (75.5) |
| Unmarried* | 216 (25.5) |
| **Education** | |
| None | 21(2.4) |
| Qur'anic | 323(36.6) |
| Primary | 56(6.3) |
| Secondary | 257(29.1) |
| Tertiary | 226(25.6) |
| **Religion** | |
| Islam | 824 (93.3) |
| Christianity | 46 (5.2) |
| Others* | 13 (1.5) |
| **Ethnicity** | |
| Hausa | 718 (81.3) |
| Fulani | 26 (2.9) |
| Yoruba | 94 (10.6) |
| Igbo | 28 (3.2) |
| Others** | 17 (1.9) |
| **Occupation** | |
| Civil servant | 147(16.6) |
| Farmer | 127(14.4) |
| Business | 301(34.1) |
| Others*** | 308(34.9) |
| **Residence** | |
| Rural | 439 (49.7) |
| Urban | 444 (50.3) |

* African tradional religion and Artheism

** other ethnic minorities

*** Artisan, Hauling, Blogging, Real estate.

In Fig 3, the major reason why respondents said they would not accept COVID-19 vaccine was that they felt they were healthy (35.2%), followed by those that said they do not think they will contract the disease (31%). A very small proportion said it was because they got some discouraging messages regarding the vaccine on social media (1.3%).

More than three-quarters 228(78.1%) of the respondents in the age group 20–29 years were willing to be vaccinated compared to three fifth 35(63.9%) of the respondents aged greater than 60 years old and the difference was statistically significant. Slightly close to three-quarters 490(73.8%) of the respondents that were married were willing to be vaccinated with covid-19 vaccines compared to slightly more than two thirds 146(67.9%) of unmarried respondents.

**Table 2. Perception of respondents regarding the COVID-19 vaccine.**

| Variable | Response | | | | | Aggregate response (Median score) |
|---|---|---|---|---|---|---|
| | SD n(%) | D n(%) | N n(%) | A n(%) | SA n(%) | |
| Enough scientific research has been conducted on the safety of the COVID-19 vaccine | 32(3.7) | 116 (13.5) | 181 (21.0) | (430 (49.9) | 102(11.8 | Agree |
| Covid 19 vaccine should be trusted if available | 62(7.1) | 86(9.8) | 136(1.5) | 419(47.7) | 176 (20.0) | Agree |
| Believe Covid-19 is man-made | 53(7.6) | 321 (46.1) | 258 (37.1) | 0(0.0) | 64(9.2) | Disagree |
| Believe Covid-19 vaccine can lead infertility | 60(7.0) | 338 (39.3) | 302 (35.1) | 112(13.0) | 49(5.7) | Neutral |
| Covid-19 vaccine is a way of implanting microchip into the body | 84(9.7) | 337 (39.0) | 223 (25.8) | 165(19.1) | 54(6.3) | Neutral |
| Covid-19 vaccine is effective in stopping Covid-19 | 39(4.6) | 257 (30.4) | 190 (22.5) | 285(33.7) | 74(8.8) | Neutral |
| There are side effects associated with the vaccine are worse than Covid-19 itself | 113 (13.2) | 353 (40.0) | 187 (21.8) | 139(16.2) | 66(7.7) | Disagree |

Key: SD = Strongly disagree, D = Disagree, N = Neutral, A = Agree, SA = Strongly agree.

More than three quarter 413(76.6%) of the respondents who had formal education compared to three fifth 223(65.6%) who had no formal education were willing to accept covid -19 vaccine. A quarter 10 (25.0%) of the respondents who had good perception compared to 626 (74.7%) who had poor perception were willing to accept the covid-19 vaccine and the difference was statistically significant (Table 4).

Factors significantly associated with perception about COVID 19 vaccine are marital status (p<0.001), ethnicity (p = 0.015) and occupation (p = 0.018). Other sociodemographic factors were not significantly associated with perception (p>0.05) (Table 5).

In Table 6, all independent variables that were found to be significantly associated with willingness to vaccinate against Covid -19 in the bivariate analysis were included in the logistic

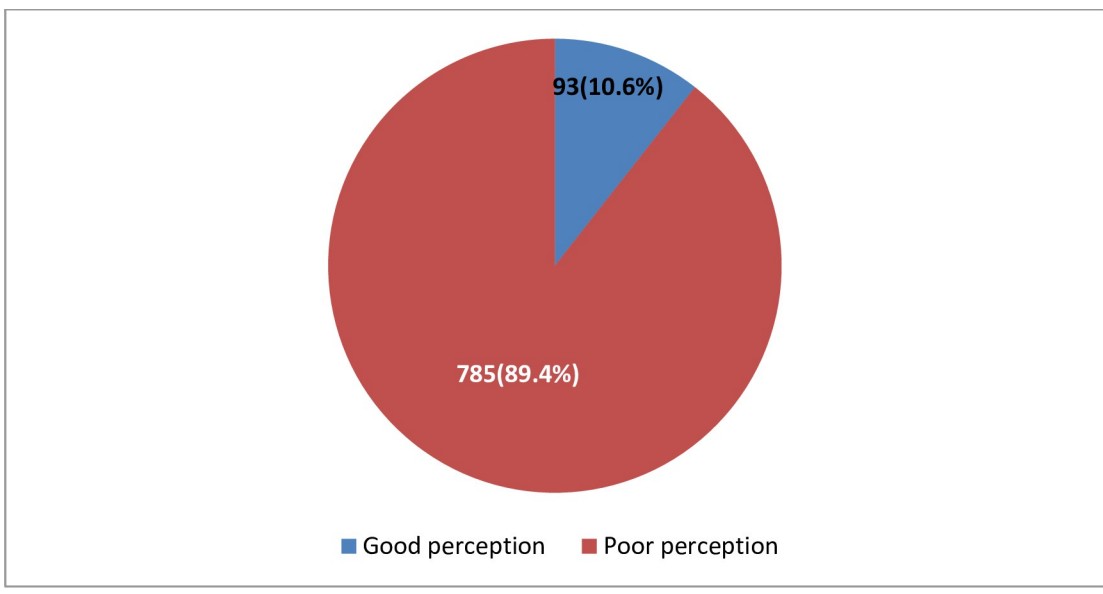

**Fig 1. Perception of respondents regarding COVID-19 vaccine.**

**Table 3. Willingness of respondents to accept COVID- 19 vaccine.**

| Variable | Response | |
|---|---|---|
| | Yes (%) | No (%) |
| Would want to be vaccinated for COVID-19 | 636(72.4) | 243(27.6) |
| Ever rejected a vaccine for my child | 148(16.8) | 731(83.2) |
| Ever decided against vaccinating myself | 264(30.0) | 615(70.0) |
| Willing to spend more than one hour in travel time to get a vaccine | 410(47.4) | 455(52.6) |
| If a new COVID-19 vaccine becomes available I will accept the vaccine for my family | 487(55.4) | 392(44.6) |

regression model; age was found to be a significant predictor of willingness to accept Covid -19 vaccine; those below the age of 20 years were about 5 times less likely to take Covid -19 vaccine compared to those who were aged 60 years and above (p = 0.006, aOR = 0.223, 95% CI = 0.077–0.645). Those with formal education almost twice more likely to accept the vaccine compared to those with non-formal education (p<0.001, OR = 1.720, 95% CI = 1.274–2.321). Those with poor perception regarding Covid -19 vaccine were up to 50 times less likely to accept Covid -19 vaccine (Table 6).

## Discussion

This study was carried out amongst the adult general population of Sokoto state to access their perception, acceptance and hesitancy to covid-19 vaccines.

The perception and willingness of the general population are a sine qua non for improving the vaccination rates and thus the herd immunity of the general population. In this study, only a few of our respondents (9.2%) believed that Covid-19 is man- made; however this is in contrast to the findings from a 15-nation study on perceptions of covid-19 where more than half (67%) of the general population of Nigeria felt that the threat from coronavirus is exaggerated and therefore could not pose any significant risk as some have suggested [33].

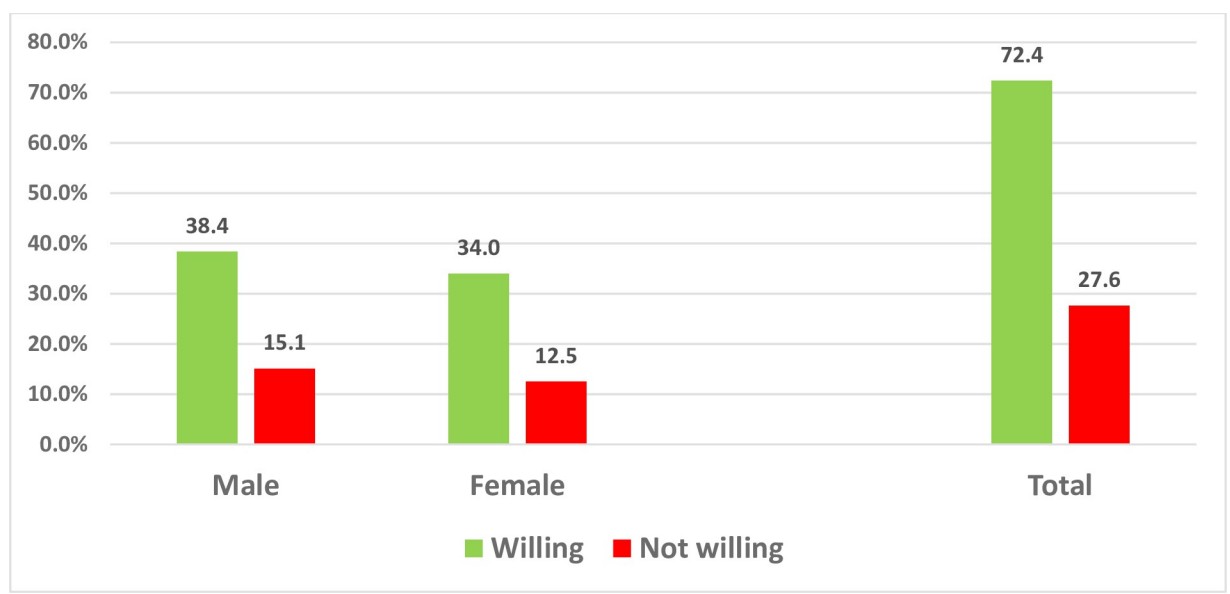

**Fig 2. Willingness to accept COVID 19 vaccine according to gender.**

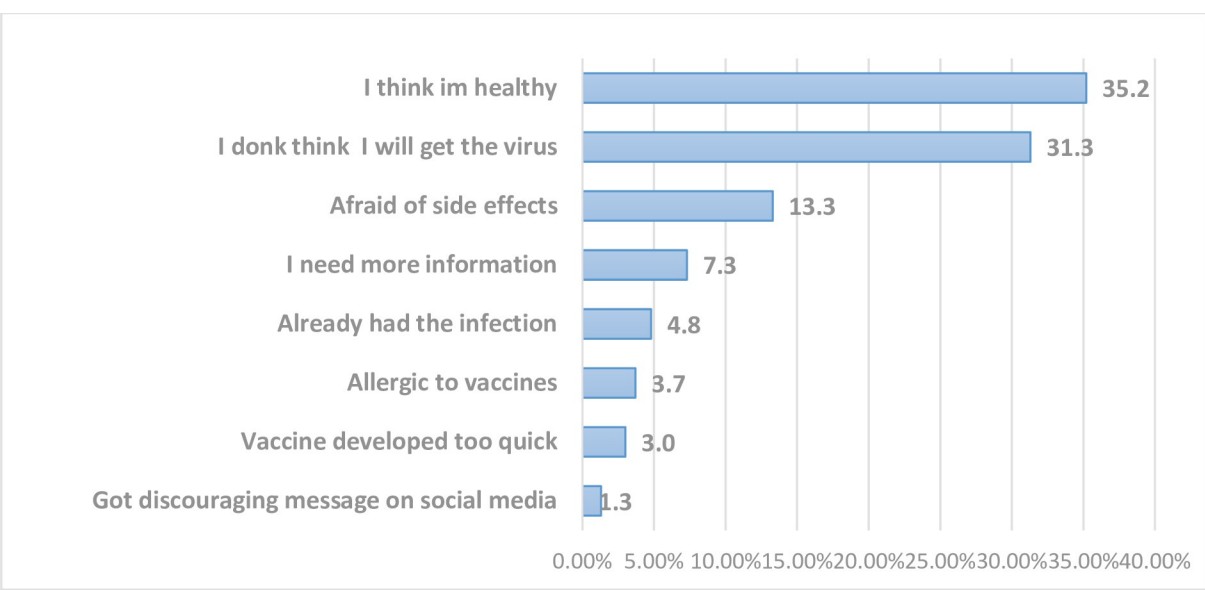

**Fig 3. Respondents' reasons for not willing to accept COVID 19 vaccines.**

Also in our study, most of the respondents disagreed that vaccines are effective against covid-19. However, in a separate multi-center study involving health workers from southern Nigeria, 66.7% of the respondents had some reservations concerning the vaccine with 43% of them believing that the vaccine might not be safe [34]. In the study by Islam and colleagues, they observed that 89% of their study subjects believed the covid-19 vaccine may have side effects [35].

Overall, 10.6% of our study subjects had good perception of the covid-19 vaccines which is not good for eventual acceptance of the vaccine. In a similar study amongst the general population in Saudi Arabia, about 71.3% of their respondents exhibited a good perception of the vaccine [36]. In contrast to these findings, Lee and his colleagues in South Korea noted a very high negative perception of the covid-19 vaccine [37]. The difference in the perception levels observed in our study and that of South Korea may not be unrelated to the fact that as of April 2021, South Korea has reported only 10,683 confirmed Covid-19 cases and 232 covid-19 related deaths [38].

It should be noted that the first case of the novel coronavirus (COVID-19) in South Korea occurred in late January 2020, approximately two months after the first case globally occurred in the Hubei province of China [39].

So far, a total of 61.8% of the world population has received at least one dose of covid-19 vaccine with 10.38 billion doses administered globally and only 10.6% of the people in low-income countries receiving one dose [40].

The increasing number of fatalities associated with the Covid-19 pandemic at its onset may not be unrelated to the non-availability of vaccines to combat the disease. To arrest the increasing morbidity and mortality due to COVID-19, researches have been conducted for the development of a COVID-19 vaccine, and COVID-19 vaccines are currently available in some countries [41]. However, because clinical trials for vaccines advanced very rapidly, and vaccines were approved in accelerated processes over a short period, negative information regarding COVID-19 vaccines has proliferated [42], due to which the number of people refusing to be vaccinated has increased.

Table 4. Factors influencing willingness to accept COVID-19 vaccine.

| VARIABLE | Willingness to vaccinate | | Test statistics | P-value |
|---|---|---|---|---|
| | Willing f (%) | Not willing f (%) | | |
| **Age (years)** | | | | |
| <40 | 45575.2) | 150(24.8) | $\chi^2 = 8.092$ | **0.017** |
| 40–59 | 146(66.7) | 73(33.3) | | |
| ≥60 | 35(63.9) | 20(36.4) | | |
| **Sex** | | | | |
| Male | 337(71.7) | 133(28.3) | $\chi^2 = 0.215$ | 0.651 |
| Female | 299(73.1) | 110(26.9) | | |
| **Marital status** | | | | |
| Married | 490(73.8) | 174(26.2) | $\chi^2 = 15.991$ | **0.001** |
| *Unmarried | 146(67.9) | 69(32.1) | | |
| **Education** | | | | |
| No formal | 223(65.6) | 117(34.4) | $\chi^2 = 27.177$ | **<0.001** |
| Formal | 413(76.6) | 126(23.4) | | |
| **Religion** | | | | |
| Islam | 593(72.3) | 227(27.7) | $\chi^2 = 0.120$ | 0.970 |
| Christianity | 34(73.9) | 12(26.1) | | |
| Others | 9(69.2) | 4(30.8) | | |
| **Ethnicity** | | | | |
| Hausa | 518(72.5) | 196(27.5) | $\chi^2 = 0.892$ | 0.927 |
| Fulani | 65(69.1) | 29(30.9) | | |
| Yoruba | 20(76.9) | 6(23.1) | | |
| Igbo | 21(75.0) | 7(25.0) | | |
| Others | 12(70.6) | 5(29.4) | | |
| **Occupation** | | | | |
| Unemployed | 199(64.6) | 109(35.4) | $\chi^2 = 37.818$ | **<0.001** |
| Employed | 437(76.5) | 134(23.5) | | |
| **Residence** | | | | |
| **Rural** | 324 (74.5) | 111 (25.5) | $\chi^2 = 1.949$ | **0.175** |
| **urban** | 312 (70.3) | 132 (29.7) | $\chi^2 = 47.235$ | **<0.001** |
| **Perception** | | | | |
| **Poor** | 626(74.7) | 212(25.3) | | |
| **Good** | 10(25.0) | 30(75.0) | | |

Pearson's chi–square test

* single, divorced or widowed.

Findings from this study showed that about 72.4% of our respondents were willing to accept the covid-19 vaccines when made available. This high acceptance rate follows the pattern obtained in some other countries such as India 70% [43], USA 70% [44], Turkey 69% [45], France 74% [46] UK and Italy 86 and 92% respectively [47].

In Africa, the lowest vaccine acceptance rate was observed in Congo 27.7% [48], while the highest was reported in South Africa where 91% of healthcare workers opined that they would accept vaccination [49]. In this study, respondents who are of younger ages (20-29years), married, employed, having good perception and formal education were more likely to accept covid-19 vaccination. Other studies have corroborated our findings that males and persons of younger ages were more likely to accept the vaccines [50–52]. In contrast to our findings, El-Elimat and colleagues observed in their study that being employed was less likely to favour acceptance of covid-19 vaccines [50].

**Table 5. Factors influencing the perception of respondents regarding the COVID-19 vaccine.**

| VARIABLE | Perception | | Test statistics | P-value |
|---|---|---|---|---|
| | Good perception f(%) | Poor perception f(%) | | |
| **Age (years)** | | | | |
| <40 | 63(10.4) | 541(89.6) | $\chi^2$ = 3.855 | 0.143 |
| 40–59 | 20(9.1) | 199(90.9) | | |
| ≥60 | 10(18.2) | 45(81.8) | | |
| **Sex** | | | | |
| Male | 59(12.6) | 411(87.4) | $\chi^2$ = 4.107 | **0.048** |
| Female | 34(8.3) | 374(91.7) | | |
| **Marital status** | | | | |
| Married | 55(8.3) | 608(91.7) | $\chi^2$ = 15.080 | <0.001 |
| Unmarried | 38(17.7) | 177(82.3) | | |
| **Education** | | | | |
| No formal | 32(9.4) | 308(90.6) | $\chi^2$ = 0.816 | 0.372 |
| Formal | 61(11.3) | 477(88.7) | | |
| **Religion** | | | | |
| Islam | 82(10,0) | 737(90.0) | FE $\chi^2$ = 4.959 | 0.061 |
| Christianity | 8(17.4) | 38(82.6) | | |
| **Ethnicity** | | | | |
| Hausa | 66(9.2) | 648(90.8) | $\chi^2$ = 12.989 | 0.015 |
| Fulani | 1(3.8) | 76(81.7) | | |
| Yoruba | 17(18.3) | 25(96.2) | | |
| Igbo | 5(17.9) | 23(82.1) | | |
| Others* | 4(23.5) | 13(76.5) | | |
| **Occupation** | 10(6.8) | 137(93.2) | $\chi^2$ = 9.992 | 0.018 |
| Civil servant | 17(13.7) | 107(86.3) | | |
| Farmer | 23(7.7) | 277(92.3) | | |
| Business | 43(14.0) | 264(83.5) | | |
| Others | | | | |
| **Place of Residence** | | | | |
| Rural | 20(4.6) | 415 (95.4) | $\chi^2$ = 32.714 | **<0.001** |
| Urban | 73(16.5) | 379(83.5) | | |

Pearson's chi–square test.

*Others (Religion = African traditional religion); (Ethnicity = Ethnic minority tribes).

In contrast to the high acceptance rate of covid-19 vaccines in our study, low rates were observed in other settings such as Lebanon 21.4% [53], Syria 35.9% [54] Jordan 37.4% [50], Delta State Nigeria 48.6% [55]and 49% in Chile [56] Findings from our study indicated that more males were willing to accept covid-19 vaccines compared to females (38.4% vs 34.0%) and this is similar to the findings from Kuwait where male subjects were more likely than female subjects to accept vaccination against COVID-19 (58.3 vs. 50.9%) [57] and also the study from low and middle-income countries [58] and elsewhere [51]. A global study observed lower odds of vaccine willingness among male participants [59]. However, women in Japan demonstrated very high vaccine hesitancy compared with men [60]. The relatively low acceptance rate in these countries could be a result of general vaccine hesitancy. With low vaccine

**Table 6. Logistic regression analysis for predictors of willingness to take COVID 19 vaccine.**

| Predictor | p-value | aOR | 95% CI for OR | |
|---|---|---|---|---|
| | | | Lower | Upper |
| Age<br><20 vs ≥60*<br>20–29 vs ≥60* | 0.006<br>0.014 | 0.223<br>0.408 | 0.077<br>0.200 | 0.645<br>0.833 |
| Form of education<br>Formal vs no formal* | <0.001 | 1.720 | 1.274 | 2.321 |
| Occupation<br>Civil servant vs Others*<br>Business vs Others* | 0.003<br><0.001 | 0.352<br>0.466 | 0.178<br>0.312 | 0.695<br>0.698 |
| Perception<br>  Poor vs Good* | <0.001 | 0.020 | 0.009 | 0.044 |

*Reference category aOR = adjusted odds ratio.

acceptance, it would be extremely difficult to manage and control the current covid-19 pandemic and by extension prolong the period of the pandemic.

High vaccination rates can ultimately lead to the achievement of herd immunity which is necessary if the devastating effects and rapid spread of covid-19 are to be nipped in the bud. This herd immunity can be achieved through a threshold range of between 50–65% [61,62].

One obvious major obstacle militating against the achievement of such a goal is believed to be vaccine hesitancy and skepticism among the populations worldwide [63–66].

Vaccine hesitancy was defined by the WHO Strategic Advisory Group of Experts (SAGE) as "a *delay in acceptance or refusal of vaccination despite the availability of vaccination services*" [67] and is a major obstacle to vaccination among the general population and health workers. The acceptance of the vaccine is instrumental to ending the pandemic, especially in the face of prevailing conspiracies and myths about the vaccine. In this study, about 27.6% of our respondents were reluctant or refused (hesitancy) to receive a covid-19 vaccine. In a similar study from rural northern Nigeria, 13% of the study subjects were reluctant to receive the vaccines [68]. Low rates of vaccine hesitancy have similarly been recorded from other studies; 32.5% in Bangladesh [69], 35% amongst the adult population in Ghana [70], 19.4 and 35.8% in South Africa and Nigeria respectively [71]. Similarly, reports from Brazil and Ecuador reported about 30% hesitancy rate [71,72].

These low vaccine hesitancy rates recorded are in support of the research done by the Wellcome Trust in the Wellcome Monitor 2018 which observed that "the Low and Middle- Income countries (LMIC), in general, had lower rates of vaccine hesitancy and, for example, had fewer safety concerns about vaccines compared to High Income Countries (HICs)"[73]. In contrast to these findings, high rates of vaccine hesitancy have been recorded in Western and Eastern Europe in addition to Russia [74]. Additionally, the Middle East has been reported to have one of the highest hesitancy rates globally with rates of 72.4, 71.6 and 35.3% in Kuwait, Jordan, and Saudi Arabia respectively [75,76]. The high hesitancy rates reported in some of these countries may not be unrelated to the lack of confidence in the safety of these vaccines when they were first rolled out.

The hesitancy associated with the covid-19 vaccines did not come as a surprise because the new mRNA-based vaccines as a novel technology were received with some skepticism since no prior experience or successes with such an approach have been reported in the past. Also, the speed of development and registration of the vaccines in record time might have been associated with the hesitancy seen with the vaccines.

Findings from our study showed that some of the reasons for hesitancy included fear of side effects, lack of knowledge of the vaccines, mistrust as a result of the speed of vaccine roll-out, belief of not being at risk of getting a covid-19 disease, and misleading media messages amongst others. Similar to our findings, several studies alluded to some of the reasons volunteered by our study subjects [56,77–79].

Given the novelty of the covid-19 vaccines and based on the theory of Diffusion of Innovation, it is envisaged that many current vaccine refusers and those undecided might accept vaccination at a later time [80,81].

## Conclusion

Encouraging the uptake of vaccines (particularly those against COVID-19) requires the understanding of people whether they are willing to be vaccinated, the reasons why they are willing or unwilling to do so, and also making available the most trusted sources of information for informed decision making.

Although the willingness of our respondents to accept the covid-19 vaccines is high, there is still a handful of them who are hesitant due mainly to safety concerns, mistrust as a result of the rapidity of vaccine roll out and the negative influence of the social and traditional media. This underscores the need for proper risk communication and also to bring on board religious and traditional leaders, healthcare providers and other civil organizations who are gatekeepers on community health issues to do away with all the conspiracy theories associated with the covid-19 vaccines if the ravaging effects of the pandemic are to be nipped in the bud.

## Supporting information

**S1 File.**
(SAV)

## Acknowledgments

Authors are thankful to the Resident Doctors in the Department of Community Medicine of Usmanu Danfodiyo University Teaching Hospital for their role as research assistants and participants for providing the information used to conduct the study.

## Author Contributions

**Conceptualization:** Oche Mansur Oche, Musa Yahaya, Asmau Mohammad Wali, Abba Aji.

**Data curation:** Abdulaziz Mohammad Danmadami.

**Formal analysis:** Habibullah Adamu, Hudu Garba Illo.

**Methodology:** Oche Mansur Oche, Abdulaziz Mohammad Danmadami.

**Software:** Hafsat Muhammad.

**Supervision:** Habibullah Adamu, Musa Yahaya, Adamu Ijapa, Asmau Mohammad Wali, Hamza Yusuf, Abba Aji.

**Validation:** Hudu Garba Illo, Adamu Ijapa, Hamza Yusuf, Hafsat Muhammad.

**Writing – original draft:** Oche Mansur Oche.

**Writing – review & editing:** Habibullah Adamu, Musa Yahaya.

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
