## [Decision Letter · Decision Letter 0]

24 May 2022

PONE-D-22-08801PERCEPTION AND WILLINGNESS TO ACCEPT  COVID-19 VACCINES: A CROSS-SECTIONAL SURVEY OF THE GENERAL POPULATION OF SOKOTO STATE, NIGERIAPLOS ONE

Dear Dr. Oche,

Thank you for submitting your manuscript to PLOS ONE. After careful consideration, we feel that it has merit but does not fully meet PLOS ONE’s publication criteria as it currently stands. Therefore, we invite you to submit a revised version of the manuscript that addresses the points raised during the review process.

We look forward to receiving your revised manuscript.

Kind regards,

Harapan Harapan, MD, PhD

Academic Editor

PLOS ONE

Journal Requirements:

2. Thank you for submitting the above manuscript to PLOS ONE. During our internal evaluation of the manuscript, we found significant text overlap between your submission and the following previously published works:

Pogue K, Jensen JL, Stancil CK, Ferguson DG, Hughes SJ, Mello EJ, Burgess R, Berges BK, Quaye A, Poole BD. Influences on Attitudes Regarding Potential COVID-19 Vaccination in the United States. Vaccines. 2020; 8(4):582. https://doi.org/10.3390/vaccines8040582

Please revise the manuscript to rephrase the duplicated text, cite your sources, and provide details as to how the current manuscript advances on previous work. Please note that further consideration is dependent on the submission of a manuscript that addresses these concerns about the overlap in text with published work.

3. Please ensure that you include a title page within your main document. We do appreciate that you have a title page document uploaded as a separate file, however, as per our author guidelines (http://journals.plos.org/plosone/s/submission-guidelines#loc-title-page) we do require this to be part of the manuscript file itself and not uploaded separately.

Authors,

There are conflicting views of reviewers on this manuscript and I will give the chance to revise the manuscript based on reviewer comments. I read this manuscript carefully and found that it should be re-written by someone expert both in the field as well as English native. I will evaluate again carefully this manuscript and see how well all the comments from reviewers are addressed. Unsatisfactory might result rejection. Master data should be provided online.

Reviewers' comments:

Reviewer's Responses to Questions

**Comments to the Author**

1. Is the manuscript technically sound, and do the data support the conclusions?

Reviewer #1: Yes

Reviewer #2: No

Reviewer #3: Partly

2. Has the statistical analysis been performed appropriately and rigorously? 

Reviewer #1: Yes

Reviewer #2: No

Reviewer #3: Yes

3. Have the authors made all data underlying the findings in their manuscript fully available?

Reviewer #1: Yes

Reviewer #2: No

Reviewer #3: Yes

4. Is the manuscript presented in an intelligible fashion and written in standard English?

Reviewer #1: Yes

Reviewer #2: No

Reviewer #3: No

5. Review Comments to the Author

Reviewer #1: Review comments

General comments

This is a very good study and well written manuscript. I commend the Authors for their good efforts in this direction.

I implore the Authors to take note of the following comments and make some amendments to the manuscript based on them.

I expect the Authors to include multivariate analysis (binary logistic regression) in the result.

Authors should ensure that their manuscript is organized according to Journal guidelines.

Abstract

Results

Why were respondents less than 18 years included in the study?

Include years after indicating the mean age of the respondents

Define how good perception was determined in the methods section of the abstract

Introduction

Second line; The origin of the virus was traced……….

Line 5, change stands to stood

Page 2, line 14, focus was on studies but only one reference was give. Change studies to study. Do the same for attitudes and practices

Always indicate Sokoto state, Nigeria so as to enhance the understanding of readers

I think the focus of the study is on willingness and not hesitancy. The aim of the study as stated towards the end of the introduction section should be reviewed

Very good introduction. My commendations to the Authors.

Materials and method

Sampling technique

Since a four stage sampling technique was used, it will be good to explain the stages in sequence. In the first stage, ……….in second stage. Also, in using a probability sampling technique, the sampling frame should be indicated.

An explanation may also be required on how the sampling interval for the systematic random sampling technique was derived

There is no indication of the number of variables used to assess perception of covid-19 vaccine

Explain also how willingness to receive COVID-19 vaccine was derived

Data analysis

I wonder why binary logistic regression analysis was not done in this study especially with such a high sample size of 883. Authors are advised to include logistic regression analysis in the result.

Ethical considerations

Include the ethical approval number from the Health Research and Ethics Committee

What information was provided to the respondents on participation in the study, confidentiality, benefits and risks associated with participation in the study?

Results

Table 1

Include sample size, (n=883) under Frequency. Create a separate column for Percent (%)

The sample size for the study as obtained for Table 1 is 883 and this is different from that indicated in the abstract, (854). This should be reconciled.

Table 4

Indicate sample size, (n=883) under willingness to vaccinate

Use variable instead of variables

Use the Chi square sign instead of Test statistics

Instead of f use N

Reduce the age groupings to three or four (three will be more meaningful)

Explain others under ethnicity and religion as footnotes

1st line, page 12. ……. and the difference in proportions was found to be statistically significant. Correct for others

Do the same corrections for Table 5

Table on factors affecting perception should come first before that for willingness

Reviewer #2: Thank you for giving me the opportunity to review this article. Find below my submission

1.The abstract is unstructured

2.The introduction is bereft of justification to the study

3.The author mentioned the prevalence of COVID 19 as at 10th of February This is an old statistics

In the methods

1.How did the authors calculate perception and hesitancy

2.Chi square alone will not do justice to this work. Linear regression will help

3.How was this patients selected

4.there is gender bias here

5.Study design, sampling technique, study population and even sample size determination are not well elaborated

In the Discussion

The authors should explain their results and give a proper critic on their findings and then compare with that of other authors

The conclusion is verbose

There is no recommendation

In the references

Please visit instruction to authors on how to cite references. This is not Vancouver

Reviewer #3: Authors surveyed the general population of the Northwestern part of Nigeria for their perception and willingness to accept COVID-19 vaccines. I find the findings reported therein are interesting that despite Nigeria is a developing nation, the willingness is high (72.4%) and comparable to India, USA, France and UK. But I have several concerns before the manuscript can be accepted for publication.

1. First of all, the typos and grammatical errors could be found almost in the entire manuscript. I understand that they are technical, but enhancing the English could be very helpful to improve the readability of the manuscript. Moreover, authors need to be consistent in using COVID-19 or covid-19 (capital or simple letters).

2. Both in the Abstract and Introduction, authors framed the narrative that developing countries tend to have low willingness to COVID-19 vaccine. However, the present findings therein suggest otherwise. In my opinion, the discussion could be made more interesting if authors could explore as to why high level of willingness was obtained in Nigeria. Authors need to be more critical as to why previous study reported low vaccine acceptance in Nigeria. Implications of these findings could be useful for other developing countries to lower vaccine hesitancy.

3. In introduction, authors mentioned about behavioral control as a means to overcome the pandemic. I strongly suggest author to include the development of oral bioavailable drugs such as molnupiravir, which is suggested could be the key for the COVID-19 pandemic. Indeed, the efficacy and safety of this drug should be further studied. Cite: Masyeni et al. J Med Virol. 2022;94(7):3006-3016 – doi: 10.1002/jmv.27730

4. Authors perhaps could discuss the type of vaccine used in Nigeria. A study found that inactivated viral vaccine has a waning efficacy. Could this be attributed to the vaccine hesitance in Nigeria? Please incorporate this study: Surawan et al. Narra J 2022; 2(1): e71-doi:10.52225/narra.v2i1.71

5. Methods should be more structured: add sub-sections (such as study design, determination of sample, questionnaire and survey, and data analysis) would be very helpful for readers.

6. Authors should be more careful when comparing their results with other previously published literatures. Such as in the second paragraph of the discussion, where authors highlighted their findings that only 9.2% believed COVID-19 was man-made. Then, authors compared the finding with the fact that 67% respondents of the previous study felt the exaggeration of COVID-19 threat. The two findings are NOT relevant, therefore not comparable. This is a major weakness of the manuscript. I strongly suggest author to make revision of the entire discussion regarding this matter.

7. “Disinformation, along with ‘haram’ notion negatively affects the COVID-19 vaccine acceptance.” This is suitable with the findings because samples with non-formal education (supposedly from Qur’anic education) is less-likely to be willing. And also cite: Hassan et al. Narra J 2021; 1(3): e 57 - doi: 10.52225/narra.v1i3.57

8. “This herd immunity can be achieved through a threshold range of between 50-65% [61,62].” The cited literatures did not consider the emergence of COVID variants. Please refer to the recent literatures such as this one: Caldwell et al. Paediatric Respiratory Reviews2021; 39: 32-39 – doi: 0.1016/j.prrv.2021.07.002

9. A study from Rosiello et al. (Narra J 2021; 1(3): e55-doi: 10.52225/narra.v1i3.55) suggests that population of North African region are still low in vaccine acceptance. Please incorporate this study in your discussion.

6. PLOS authors have the option to publish the peer review history of their article (what does this mean?). If published, this will include your full peer review and any attached files.

Reviewer #2: No

Reviewer #3: No

---

## [Author Response · Author response to Decision Letter 0]

13 Sep 2022

A rebuttal letter responding to all the queries raised by reviewers have been included

All corrections and additional information requested have been made through the tracking system

A clean copy of the final revised manuscript is also enclosed

---

## [Decision Letter · Decision Letter 1]

28 Oct 2022

PONE-D-22-08801R1PERCEPTION AND WILLINGNESS TO ACCEPT  COVID-19 VACCINES: A CROSS-SECTIONAL SURVEY OF THE GENERAL POPULATION OF SOKOTO STATE, NIGERIAPLOS ONE

Dear Dr. Oche,

Thank you for submitting your manuscript to PLOS ONE. After careful consideration, we feel that it has merit but does not fully meet PLOS ONE’s publication criteria as it currently stands. Therefore, we invite you to submit a revised version of the manuscript that addresses the points raised during the review process.

We look forward to receiving your revised manuscript.

Kind regards,

Harapan Harapan, MD, PhD

Academic Editor

PLOS ONE

Journal Requirements:

Reviewers' comments:

Reviewer's Responses to Questions

**Comments to the Author**

1. If the authors have adequately addressed your comments raised in a previous round of review and you feel that this manuscript is now acceptable for publication, you may indicate that here to bypass the “Comments to the Author” section, enter your conflict of interest statement in the “Confidential to Editor” section, and submit your "Accept" recommendation.

Reviewer #1: (No Response)

Reviewer #3: All comments have been addressed

2. Is the manuscript technically sound, and do the data support the conclusions?

Reviewer #1: Yes

Reviewer #3: Yes

3. Has the statistical analysis been performed appropriately and rigorously? 

Reviewer #1: No

Reviewer #3: Yes

4. Have the authors made all data underlying the findings in their manuscript fully available?

Reviewer #1: Yes

Reviewer #3: Yes

5. Is the manuscript presented in an intelligible fashion and written in standard English?

Reviewer #1: Yes

Reviewer #3: Yes

6. Review Comments to the Author

Reviewer #1: (No Response)

Reviewer #3: Authors may consider some minor technical issues before publication which could be done during proofreading process:

a. Please be mindful on the consistent use of COVID-19 vs Covid-19. Errors in punctuations still could be found; please check and revise.

b. Table 6. The OR and 95% CI columns can be combined >> OR (95% CI)

7. PLOS authors have the option to publish the peer review history of their article (what does this mean?). If published, this will include your full peer review and any attached files.

Reviewer #1: **Yes: **EDMUND NDUDI OSSAI

Reviewer #3: **Yes: **Muhammad Iqhrammullah

---

## [Author Response · Author response to Decision Letter 1]

14 Nov 2022

These have been included in the response to reviewer Query

---

## [Editor Report · Decision Letter 2]

15 Nov 2022

PERCEPTION AND WILLINGNESS TO ACCEPT  COVID-19 VACCINES: A CROSS-SECTIONAL SURVEY OF THE GENERAL POPULATION OF SOKOTO STATE, NIGERIA

PONE-D-22-08801R2

Dear Dr. Oche,

We’re pleased to inform you that your manuscript has been judged scientifically suitable for publication and will be formally accepted for publication once it meets all outstanding technical requirements.

Kind regards,

Harapan Harapan, MD, PhD

Academic Editor

PLOS ONE
---

## [Editor Report · Acceptance letter]

22 Nov 2022

PONE-D-22-08801R2 

Perception and willingness to accept  COVID-19 Vaccines: A Cross-Sectional Survey of the general population of Sokoto State, Nigeria 

Dear Dr. Oche:

I'm pleased to inform you that your manuscript has been deemed suitable for publication in PLOS ONE. Congratulations! Your manuscript is now with our production department. 

Kind regards, 

on behalf of

Dr. Harapan Harapan 

Academic Editor

PLOS ONE